# Role of Polyamines in the Response to Salt Stress of Tomato

**DOI:** 10.3390/plants12091855

**Published:** 2023-04-30

**Authors:** Ilaria Borromeo, Fabio Domenici, Maddalena Del Gallo, Cinzia Forni

**Affiliations:** 1PhD Program in Evolutionary Biology and Ecology, Department of Biology, University of Rome Tor Vergata, 00133 Rome, Italy; ilaria18scv@hotmail.it; 2Department of Chemical Science and Technologies, University of Rome Tor Vergata, Via della Ricerca Scientifica, 00133 Rome, Italy; fabio.domenici@uniroma2.it; 3Department of Health, Life and Environmental Sciences, University of L’Aquila, Via Vetoio, Coppito 1, 67100 L’Aquila, Italy; maddalena.delgallo@univaq.it; 4Department of Biology, University of Rome Tor Vergata, Via della Ricerca Scientifica, 00133 Rome, Italy

**Keywords:** acclimation, polyamines, saline water, seed priming, tomato

## Abstract

Plants irrigated with saline solutions undergo osmotic and oxidative stresses, which affect their growth, photosynthetic activity and yield. Therefore, the use of saline water for irrigation, in addition to the increasing soil salinity, is one of the major threats to crop productivity worldwide. Plant tolerance to stressful conditions can be improved using different strategies, i.e., seed priming and acclimation, which elicit morphological and biochemical responses to overcome stress. In this work, we evaluated the combined effect of priming and acclimation on salt stress response of a tomato cultivar (*Solanum lycopersicum* L.), very sensitive to salinity. Chemical priming of seeds was performed by treating seeds with polyamines (PAs): 2.5 mM putrescine (PUT), 2.5 mM spermine (SPM) and 2.5 mM spermidine (SPD). Germinated seeds of primed and non-primed (controls) were sown in non-saline soil. The acclimation consisted of irrigating the seedlings for 2 weeks with tap water, followed by irrigation with saline and non-saline water for 4 weeks. At the end of the growth period, morphological, physiological and biochemical parameters were determined. The positive effects of combined treatments were evident, when primed plants were compared to non-primed, grown under the same conditions. Priming with PAs improved tolerance to salt stress, reduced the negative effects of salinity on growth, improved membrane integrity, and increased photosynthetic pigments, proline and enzymatic and non-enzymatic antioxidant responses in all salt-exposed plants. These results may open new perspectives and strategies to increase tolerance to salt stress in sensitive species, such as tomato.

## 1. Introduction

Lack of rainfall, high temperatures and poor snowfall during the winter season are the main factors that lead to drought related to the climate crisis, with dramatic consequences for the irrigation of fields and for the supply of drinking water. This scenario is even more complicated by the limited water availability and enhancement of salts in the water, due to the fact the seas are flowing back into the branches of river deltas and, in coastal areas, infiltrating increasingly low aquifers due to excessive drawing from irrigation wells [1]. Irrigation water is one of the main factors affecting crop yield and agricultural production. It has been estimated that agricultural water use represents about 70% of global freshwater extraction and consumption; however, arid and semi-arid areas are characterized by severe freshwater shortage. Since most water supplies are preferentially destined for urban uses, the amount of freshwater available for agricultural irrigation is extremely limited [2,3]. We are aware of the negative impact on plant growth and productivity of water deficiency, and nowadays different research efforts are focused on the development of strategies to mitigate its effects on crop yield by selection of cultivars more tolerant to drought and/or by improving soil management and irrigation techniques. Proper irrigation is essential for crop growth and yield because water is an important constituent of plant cells and plays a key role in many biochemical processes [4]. Due to the limited amount of fresh water that can be used for crop irrigation in arid and semi-arid areas, it will be necessary to develop water management strategies, such as drip irrigation or use of low-quality water, suitable for obtaining the highest yield from crops [5,6]. In areas suffering from freshwater shortage, irrigation with saline water can partially satisfy the plants’ demand for water and maintain a certain level of crop yield; therefore, it is very important to develop agricultural management practices to overcome the effect of salinity, and optimize the use of irrigation with saline water [5]. On the other hand, although irrigation with saline or poor-quality water partially relieves the problems related to water shortages, it can cause other problems. It is well known that continuous irrigation with saline water, compared to irrigation with fresh water, leads to soil salinization, reducing crop growth and agricultural productivity [2]. This is due to the presence of salt in the root zone, which limits the uptake of water and nutrients by the roots, inhibiting plant growth. In fact, a high concentration of salts in the root zone prevents the flow of water from the soil to the epigeal parts of the plant, while the accumulation of harmful ions, such as sodium and chlorine, can inhibit protein synthesis and inactivate various enzymes [7]. Moreover, continuous irrigation with saline water can lead to secondary salinization of cultivated fields, with serious consequences for the agricultural environment and the ecosystem [8]. Thus, the effectiveness of irrigation with saline water is very limited, because the concentration of sodium chloride, dissolved in the water, can significantly reduce plant growth when water salinity exceeds the salt tolerance threshold of the plant [9]; therefore, irrigation with saline water is only recommended for moderately salt-tolerant crops, while, irrigation with fresh water, in the early sensitive stages, combined with saline water in the later tolerant stages, can minimize yield loss in sensitive or moderately sensitive species [10,11,12]. Among the latter is tomato (*Solanum lycopersicum* L.), in which a reduction of yield is associated with irrigation with saline water [10,12]. High concentrations of Na^+^ and Cl^-^ reduce the water potential, which decreases water uptake by roots and, consequently, photosynthesis [13].

Worldwide, the tomato is grown in open fields, greenhouses and indoors and can be cultivated on a wide range of well-drained soils. The crop requires a lot of water for plant growth, i.e., under ideal soil moisture conditions, so that they can continue to flower, although high humidity leads to higher incidence of pests and diseases and fruit rot. Dry climates are therefore preferred for tomato production. Water requirements are related to evapotranspiration (Eto) and to the development stage, mid-season, late-season and harvesting phases (FAO, https://www.fao.org/land-water/databases-and-software/crop-information/tomato (accessed on 10 March 2023)).

This horticultural crop is moderately sensitive to salt [14] and requires constant and regular irrigation and adequate nutrients for optimal growth and production in many countries; soil and groundwater salinity are considered the major problems for tomato production in several of these countries, such as in the Mediterranean area. The range in soil electrical conductivity (EC) threshold is 1.7–2.5 dS/m, while the salinity threshold of irrigation water is 1.7 dS/m [8]; however, many studies report considerable variability of the response to salinity levels among tomato cultivars [15].

Cultivation problems occur during the dry season or when freshwater for irrigation is not available, as crop performance is sensitive and linked to irrigation practice: prolonged and severe water deficit not only limits growth, but also affects the yield (FAO). In fact, yield decreases at various electrical conductivity (EC) values have been reported (FAO, https://www.fao.org/land-water/databases-and-software/crop-information/tomato (accessed on 10 March 2023)).

Sensitivity to salt is related to developmental stage; i.e., the most sensitive period is during germination and early plant development (FAO, https://www.fao.org/land-water/databases-and-software/crop-information/tomato (accessed on 10 March 2023)). Thus, the irrigation regime plays a vital role for this crop and should be adjusted according to the stage of development of the plants.

The salinity of irrigation water affects the vegetative growth of tomatoes, by reducing stem and roots length, leaf area, and dry weight of plants [16,17,18,19,20,21,22]. Conversely, a beneficial effect of salinity on tomato fruit quality has been reported, i.e., higher levels of salt improve quality by increasing fruit taste, sugar content and total soluble solids, without affecting shelf life [16]. Less salt-sensitive wild tomato species (e.g., *Solanum pimpinellifolium*) are known to show adaptations in salt accumulation and expression of genes involved in sodium transport, salicylic acid signaling and detoxification of sodium chloride [19,21]. Little information has been reported on the influence of NaCl during the first weeks of growth of cherry tomatoes grown in greenhouses, and on possible strategies to increase their yield, avoiding damage caused over time by continuous irrigation with strongly saline water.

Several approaches, besides breeding and genetic engineering, are thought to improve crop tolerance to abiotic stresses. Seed priming is a pre-sowing treatment, which consists of soaking the seeds in a priming agent, followed by drying them to avoid rootlet emergence. Interestingly, during subsequent germination, primed seeds show a faster and more synchronized germination, followed by more vigorous seedling development with respect to non-primed seedlings [23]. In addition, a priming agent can provoke abiotic stress in the seed, inducing a cross-tolerance to different abiotic stresses [24,25,26]; therefore, the priming process induces a generally better tolerance to abiotic stresses, as reported by several authors [23,25,26]. This approach has been successfully applied to improve stress tolerance in several crops [27].

On the other hand, an acclimation to stress can be obtained through a gradual exposure of the plant to stressful conditions, leading to a better plant adaptation to stress, as a result of physiological, biochemical and molecular adjustments switched on within cells [24,28]. For example, salt acclimation can enhance the tolerance of rapeseed plants to salt stress through the enhancement of antioxidant activity and the positive protection of photosynthesis [28].

Literature reports have highlighted that polyamine (PAs) levels are one of the most remarkable metabolic hallmarks in plants exposed to different abiotic stress conditions (e.g., drought, salinity, chilling, heat, hypoxia, ozone, UV, heavy metals and herbicides) [29]. PAs have been proved to be beneficial for protein homeostasis, detoxification of reactive oxygen species (ROS), activation of the antioxidative machinery and molecular chaperone activity under stress conditions [26,29]. Exogenous application of PAs has been successful in ameliorating abiotic stress tolerances in different species [26,30,31].

Based on the economic importance of tomatoes, an experimental greenhouse trial was conducted on salt-sensitive cherry tomato (cv. Principe Borghese), to determine the effects of seed priming with PAs on acclimation to salt.

## 2. Results

### 2.1. Determination of Halotolerance, Priming Agents and Priming Memory

Tomato is a crop sensitive to salt stress during the germination phase. To determine the halotolerance of the cv. Principe Borghese, a seed germination test was performed (Figure 1). Seeds placed on filter paper, imbedded with 15 mL of saline solution, showed a delay in germination related to NaCl concentration. After 7 days, seeds exposed to 80 mM NaCl (threshold of salt sensitivity) showed a significant reduction in germination rate, while germination was totally inhibited by160 mM NaCl.

To improve plant tolerance to salt, different concentrations of various priming agents were tested. The germination rate was recorded every 24 h for 5 days. The best priming agents were: 2.5 mM PUT, 2.5 mM SPM, 2.5 mM SPD. Seed priming led to a significant increase in germination during the first 72 h; no priming agent led to a marked reduction of germination rate after 5 days (Table 1).

To check whether the seeds maintained a priming memory over time, a final germination test was carried out one year after the priming treatment. The seeds, primed with PUT and SPD, germinated more than the others, even one year after the priming treatment. After 7 days, the germination rate of all primed seeds was significantly higher than that of the control seeds (Table 2). Maintenance of priming memory was further confirmed when the plants, developed from these primed seeds, and irrigated with saline solutions, were still showing better salt tolerance than the controls (Appendix A).

### 2.2. Experiments on Soil

Experiments were carried out in vivo by directly sowing the seeds on saline soil (EC = 4.2 dS/m). Although priming did not affect seed germination, primed and germinated seeds did not survive when sown in saline soil. To overcome this constraint, we planned new experiments, where we applied the combination of priming and acclimation. Accordingly, the primed seeds were sown in non-saline soil, acclimated for 2 weeks, and then irrigated gradually, twice a week, with saline water.

Irrigation with saline solutions led to a gradual increase of soil salinity; EC were increased from 16% up to 30% in pots irrigated with the highest salt concentration compared to controls (Table 3). Due to the higher water uptake by primed plants, soils irrigated with 0 and 80 mM NaCl showed a significant decrease in gravimetric water content (GWC) compared to the relative CTRL (Table 3).

### 2.3. Effects of Combined Treatments on Plants Growth and Chlorophylls

At the end of the experiments, the growth of primed plants was compared to non-primed (controls), exposed or not to salt after acclimation. In control plants, the increased salinity reduced both shoot and root length (Table 4), plant water content (PWC) and salt tolerance index (TI) (Table 4). A significant reduction of the negative effect of salt on growth was found in plants primed with all Pas; furthermore, PUT and SPD also had a positive effect on emergence of new leaves (Figure 2, Table 4).

The positive effect of the priming and acclimation treatments was even more pronounced in the roots: the significant growth decrease, caused by the salt and determined in non-primed plants, was counteracted in primed plants (Table 4). Besides, an increase of the salt tolerance index (TI) was observed in the latter, particularly in plants irrigated with 160 mM NaCl solutions. As expected, a decrease in TI was detected in controls exposed to high salinity. Water content was not significantly different in primed and non-primed plants irrigated with saline solutions (Table 4).

In general, PA priming had significant positive effects on chlorophylls in plants exposed or not exposed to salt (Table 5). Vice versa, salty water led to a decrease of chlorophylls in the controls.

### 2.4. Changes in Metabolism: Soluble Sugars, Phenolic Compounds and Proline

Salt exposure significantly increased soluble sugars content in the controls. While a significant decrease in their concentration was found in plants primed with SPM and SPD and irrigated with saline solutions, no significant differences were found between the levels of salinity, whereas no change was observed in plants primed with PUT (Table 6).

Proline may offset cellular imbalances caused by various environmental stresses, thus the use of proline as a stress adapter molecule indicates its critical role in stress response. Seed priming enhanced the amount of proline (Table 6).

In SPD primed plants, proline synthesis already showed significative enhancement at 80 mM NaCl (+57%) (Table 6). Under salt stress, PUT was the polyamine that increased proline production (+128%) more than any other (Table 6).

Exposure to saline conditions can induce changes in the metabolism of secondary compounds. In this work, we detected significantly lower amounts of phenolic molecules in non-primed controls under saline conditions. Synthesis was significantly enhanced in primed and exposed plants compared to controls (Table 7).

A similar trend was observed in flavonoids synthesis in non-primed and primed plants with PUT and SPM, while in SPD samples salt exposure did not increase flavonoid production (Table 7).

### 2.5. Lipid Peroxidation Inhibition

Through the analysis of thio-barbituric acid (TBA) reactive products, it was possible to determine the degree of membrane lipid peroxidation, expressed as mmol of malondialdehyde (MDA) equivalent/gram fresh weight. A higher MDA production suggests a higher level of lipid peroxidation, thus elevated membrane damage. Except for PUT, salt exposure increased lipid peroxidation (Figure 3).

### 2.6. Antioxidant Activity

One of the main effects of plant exposure to salt stress is the overproduction of ROS. To reduce the negative effects of ROS on cells, plants increase antioxidant activity by enhancing the production of antioxidant molecules and the activity of many enzymes. Antioxidant activity was determined in samples of treated and untreated plants. The saline treatment led to a significant reduction of antioxidant activity (AA) in the control extracts. A decrease in AA was also found in unstressed primed plants, whereas plants primed with PUT and SPM exhibited a significant increase in antioxidant power (+17% and +14% respectively), after irrigation with 160 mM saline solution (Figure 4).

As shown in Figure 5, the control plants showed a marked decrease in reducing power (RP) after irrigation with 160 mM NaCl (−45%) compared to CTRL (0 mM NaCl). All extracts from primed plants, irrigated with 160 mM NaCl, showed a marked increase in RP (+100%, +80% and +90% with PUT, SPM and SPD, respectively) (Figure 5).

Similar results were obtained using the PFRAP test (Figure 6): control plants had a reduction in RP only at the highest salt irrigation (−10%); a completely opposite trend was observed in primed plants. These results suggest that only strongly saline solutions elicited RP response.

The enzymatic response was evaluated by determining the activity of the following enzymes: peroxidase (POD), superoxide dismutase (SOD), polyphenol oxidase (PPO) and ascorbate peroxidase (APX). In all primed plants, POD and PPO activities were increased in both not saline and saline conditions (Figure 7 and Figure 8). SPM and SPD improved POD activity under strong saline irrigation (+900% and +800% respectively), while PUT was the polyamine mostly associated with an increase in PPO activity, especially under strong saline irrigation (+980%) (Figure 8).

SOD activity showed a different trend respect to POD and PPO: the most important results were observed with PUT priming (+11% and +18% respectively at 80 and 160 mM NaCl); all plants showed an increase in its activity in both growth conditions (Figure 9a,b). A slight increase of the activity was detected in SPM (+12%) and SPD (+13%) primed plants irrigated with 160 mM NaCl (Figure 9c).

APX responded to saline conditions in both controls and primed plants (Figure 10). The latter showed an upregulation of the activity of the enzyme, even in not saline conditions (Figure 10).

### 2.7. Relationship between Calcium and the Enzyme Transglutaminase (TGase)

TGases have been reported as being associated with several stages of plant development and responses to biotic and abiotic stresses. Concerning calcium levels and TGase activity, a different behavior was observed in shoots and roots of control plants. In the latter exposed to salt, calcium was mostly accumulated in roots (Table 8). Shoots of primed plants had higher calcium levels that was related to soil salt concentration (Table 8); while in roots, a marked decrease of calcium was detected (Table 8) in both not-stressed and stressed primed plants. 

Similarly to calcium levels, an increase of TGase activity was detected and related to salinity, being most elevated in shoots than in roots. TGase activity of shoots and roots was specular to calcium concentration in these organs (Figure 11a,b).

## 3. Discussion

In many areas, climate change will lead to reduced availability of water for agriculture, therefore accurate and careful agronomic practice will be needed. It is well known that agricultural practices consume more than 85% of available freshwater and 80% of the annual phosphate rock extracted globally [32]. The growing demand for high quality water, for both domestic and industrial purposes, forces the use of non-conventional water resources for irrigation [6]. Whenever freshwater is limited, drip irrigation and/or poor-quality water utilization have been considered as an alternative for irrigation in agriculture [6,29,33]. Based on the possible negative effects on plant growth, the use of marginal quality water for crop production should be carefully considered, and treated prior to application in the crop field. Furthermore, suitable agronomic techniques are needed and the selection of stress tolerant crops or varieties are necessary to avoid the loss of soil fertility. The latter can be the consequence of the accumulation of salts in the soil, leading to enhancement of salinity [33].

The accumulation of salts causes toxicity to plants, especially when grown in pots, but even in open fields; therefore, the risk of progressive soil salinization should not be underestimated. Soil deterioration is certainly the best-known negative effect of this irrigation practice, showing a direct effect on agricultural soils. The water lost, through evapotranspiration, is essentially pure, thus the salts, present in traces in such irrigation water, will be concentrated in the soil. Continuous irrigation and the progressive accumulation of these salts will elicit osmotic imbalance, and the toxic effect of certain ions on crops could be so severe as to reach soil sterility.

Unfortunately, the areas affected by soil salinization are increasing, especially in arid and semi-arid regions, where high evapotranspiration fluxes are combined with low rainfall. Plant species are characterized by a different degree of tolerance to salinity and divided into four classes: tolerant, moderately tolerant, moderately sensitive and sensitive. Irrigation with saline water and the subsequent salinization of the soil strongly affect crop growth, causing yield loss. To improve crop tolerance to salt stress, several strategies are adopted, including genetic engineering, plant acclimation and seed priming [26,28].

PAs are molecules involved in several stages of plant growth and development, as well as in responses to biotic and abiotic stresses [26,34]. Several authors have reported the relationship between PA metabolism and plant responses to nutrient deficiency, low or high temperature stress, drought, salinity, heavy metal excess and pesticide treatments [26,27,29,35].

In the present work, we analyzed the effects of PUT, SPD and SPM, as priming agents, on salt stress response of the sensitive tomato cv. Principe Borghese. Seed priming treatment with PAs was not toxic, and even increased seed germination in saline conditions. In the first 72 h, faster germination was observed in the primed seed than in the control. Subsequently, based on these encouraging results, we tested salt tolerance of tomato grown in saline soil (EC = 4.2 dS/m), and we found that priming per se was not sufficient to support growth. It was clear that seed priming and acclimation, if used separately, were not able to increase plant tolerance to salt stress; thus, a combination of both techniques was necessary to create a new protocol that allows growth under stress conditions.

The performance of primed and acclimated plants was better than that of controls: higher growth rates and chlorophyll content were detected in more saline soils. The tolerance index revealed an improved salt tolerance by means of priming with all PAs tested, in contrast to another study, where only PUT (1 mM) enhanced plant growth [36]. According to Hu et al. [37], in tomato roots exogenous SPD, applied as a pre-soaking treatment to seeds, promoted conversion of free PUT into free SPD and SPM as well as soluble conjugated and insoluble bound Pas, further enhanced ODC, SAMDC, PAO and DAO activities, and enhanced tolerance of tomato plants to salinity.

There are many controversies about the effect of salinity on the amount of chlorophyll. Many researchers have reported that salinity reduces chlorophyll in salt-sensitive plants and increases it in salt-tolerant plants [38,39,40]. Our data agree with those showing a decrease of chlorophyll in plants exposed to salt stress and non-primed [38], while priming treatment with PAs improved the tolerance of tomato plants to salt stress with respect to chlorophyll maintenance.

Changes in carbohydrates are important for their relationship with physiological processes, such as photosynthesis; in fact, there is a close relationship between chlorophyll and soluble reducing sugars in plants [41,42,43,44,45]. The increase in the content of sugars has been observed in many plant species exposed to salinity. This phenomenon contributes to an osmotic regulation, allowing plants to optimize the resources required to maintain basal metabolism in a stressful environment [41,45].

The expression of Rubisco can be repressed by an excess of sugars in the cytoplasm. A reduction in photosynthesis and metabolic changes, due to the accumulation of sugars, could contribute to salt sensitivity, which inhibits the growth of the salt-sensitive cultivar under salt stress [42]. The results seem to support the literature data: in stressed controls, the carbohydrate level was significantly higher, when compared to primed plants, with SPM and SPD.

Priming treatments, especially when SPM and PUT were used, mitigated the symptoms of membrane damages, counteracting the effect of overproduction of ROS induced by salinity [46]. Our results show that seed priming with PAs play a key role in tomato salt stress tolerance, thus modulating and increasing free radical scavenging and antioxidant activity in primed plants exposed to saline conditions. To fight and mitigate the damage caused by the overproduction of ROS due to salt stress, plants must increase the presence of non-enzymatic (phenols, flavonoids, etc.) and enzymatic (superoxide dismutase, ascorbate peroxidase, etc.) protectors. Combining the results of DPPH, FRAP and PFRAP assays, it was observed that PAs enhanced antioxidant activity, i.e., a significant increase in reducing power and scavenger activity was recorded in primed plants exposed to high salt level (160 mM NaCl).

PAs might be positive activators of non-enzymatic protective pathways, reducing the production of radical species, such as DPPH, ferric tripyridyl-triazine or potassium ferrocyanide. To increase antioxidant activity, plants can also enhance the synthesis of osmolytes, such as proline, and secondary metabolites, such as phenols and flavonoids [26,39,40], which are species-specific and associated with plant protection against both abiotic and biotic stresses. Although the synthesis of secondary metabolites increases when plants are grown under salt stress conditions, many studies pay particular attention to only a few target compounds.

Proline acts in osmotic regulation, but it is also a reserve of energy and nitrogen, to be used in stress conditions; while phenols and flavonoids are the main antioxidant secondary metabolites, providing another protection against the formation of radical species. An accumulation of both proline and secondary metabolites was observed in almost all primed plants, in agreement with the literature [26,39]. These data suggest an involvement of PAs in the synthesis of these metabolites, in a related manner to the increase of salinity. These data agree with those of Aziz et al. [47], who reported a correlation between proline accumulation and salt tolerance in *Lycopersicon esculentum* and *Aegiceras corniculatum.*

The overproduction of ROS, caused by salt stress, can be balanced by antioxidant enzymatic activity [48], which involves several enzymes, such as superoxide dismutase (SOD), peroxidase (POD) and ascorbate peroxidase (APX). Studies have shown that, under saline stress, their activities are increased in salt-tolerant plants, but decreased in salt-sensitive ones [26,28]. Polyamine accumulation in plants under salt stress helps maintain cellular ROS homeostasis. The role of PAs in mediating responses to salt stress through their modulation of redox homeostasis has been outlined in a review by [49], although there are conflicting reports in the literature, where the increase in cellular PAs levels during salinity stress can exert a dual effect. On the one hand, exogenous polyamine application was correlated with higher plant tolerance to abiotic stress, partly due to the increased ability to inactivate oxidative radicals. On the contrary, [35] it has been reported that PAs decrease plant’s ability to cope with stress, possibly because of the increased levels of H_2_O_2_, derived from their catabolism. Consequently, in stressed plants, Pas’ role could be quite complex to elucidate.

In our study, we detected a general positive effect on enzymatic activity in primed plants, such as an increase in PPO activity, related to salinity and the synthesis of phenolic compounds, or both POD and APX activity, as reported for salt-tolerant plants in various studies [28,39,50]. Therefore, PAs are activators of non-enzymatic and enzymatic antioxidant responses, with marked differences, depending on the PAs used as priming agent. SPM was the PA most involved in the activation of peroxidases: an increase in POD corresponds to an up-regulation of APX, both observed in samples treated with SPM. Instead, PUT and SPD are considered as activators of SOD.

Salt exposure causes an accumulation of Na^+^ and Cl^-^ in shoots, which leads to ionic stress with loss of K^+^, Ca^2+^ and Mg^2+^. This ionic imbalance prevents stem elongation and the production of new leaves. In plants, Ca^2+^ ion behaves as a crucial second messenger in signaling pathways coupling the perception of environmental changes to plant adaptive responses. Indeed, the intracellular variation of free calcium concentrations represents one of the earliest events and key point following the perception of stress [51]. As reported by [52], salinity stress inhibits Ca^2+^ translocation in the shoot, leading to Ca^2+^ deficiency. Ion transport is inhibited in salinized roots due to osmotic stress.

Shoots of not-primed plants, irrigated with saline solutions, had significantly lower amounts of Ca^2+^ than stressed primed plants. Vice versa, in roots of not-primed plants, Ca^2+^ level was significantly higher than in all primed plants. PAs increased Ca^2+^ translocation from the roots to the shoots, providing faster osmotic regulation and activation of salt adaptation responses. The higher concentration of Ca^2+^ in the stem increased antioxidant responses, supporting what was observed by Halperin et al. [52].

The increase of cytosolic Ca^2+^ increases TGase activity, leading to a better protein cross-link between PAs and cytoskeleton filaments, remodeling the cell’s response to stress, resulting in a better structural stability [53]. Interestingly, TGase appears to play a role in acclimation to high salinity levels and may regulate the post-translational modification of those proteins involved in plant responses to different environmental stresses [54], thus playing a positive role in tolerance. TGase-improved photosynthetic capacity seems to be supported by changes in the cellular redox status and activation of antioxidant enzymes [55]. For example, in the microalga *Dunaliella salina*, as reported by Parrotta et al. [54], hyper-saline stress caused a change in chloroplast TGase concentration, affecting the enzyme activity. This was proved in a PA deficient variant *Dunaliella,* showing low TGase activity, and the variant was more severely affected by salt stress; nevertheless, application of exogenous PUT resulted in a recovery of TGase activity and increased chlorophyll content. TGase-deficient tomatoes showed a decrease in the activity of antioxidant enzymes involved in the ascorbate cycle, while plants, over-expressing TGase, showed an increase in the activity of APX and other enzymes. This antioxidant machinery may prevent the imbalance in redox homeostasis caused by the excessive accumulation of reactive oxygen and nitrogen species under stress conditions [54,55]. These studies support what was observed in our treated plants. An increase in TGase was detected in the stems of primed plants, correlated with enhanced activity of APX and other antioxidant enzymes (POD, SOD, etc.) together with marked accumulation of photosynthetic pigments. PAs increased TGase activity, enhancing adaptation and tolerance responses to salt stress.

Last, but not least, the study on persistence of priming memory is currently in progress, and very few researchers have explored this aspect of priming. There is no scientific evidence to claim how long the seeds maintain priming memory. In this study, primed seeds appear to germinate more than non-primed seeds, even one year after the original treatment. Experiments verified the maintenance of the positive effect of priming during the developmental phases of the plants.

## 4. Materials and Methods

The reagents were analytical grade or equivalent and purchased from Merck or Sigma-Aldrich unless otherwise stated. In each set of experiments, all working solutions were prepared immediately prior to use from stock reagents.

### 4.1. Determination of Halotolerance

The halotolerance of the seeds was determined through a dose-response curve. Ten seeds were placed in Petri dishes, on filter paper, imbibed with 15 mL of water or salt solutions at various concentrations. Germination rates were recorded every 24 h, for 7 days. The concentration of NaCl that significantly decreased the germination of the seeds was considered as the threshold of salt tolerance.

### 4.2. Plants’ Growth Conditions and Saline Treatments

Seeds of *Solanum lycopersicum* L., cv. Principe Borghese, were chosen (Blumen Group S.p.A, Piacenza, Italy). Seeds were stored in the dark at room temperature until the priming treatment. Before priming, the seeds were surface sterilized (70% ethanol for 2 min and then soaked in a solution of 1% NaClO for 5 min) and rinsed in double distilled water.

To determine the best priming agent, preliminary tests were carried out using different priming agents at various concentrations: 40 mM NaCl (halopriming), 2.5 mM Ca(NO_3_)_2_ (osmo-priming), 2.5 mM or 5 mM of PUT, SPM, SPD (chemical priming). Based on the germination (%) of the seeds, PAs (2.5 mM) were found to be the best priming agents.

We primed the seeds with 20 mL of the following compounds: putrescine (2.5 mM PUT), spermine (2.5 mM SPM) and spermidine (2.5 mM SPD) for 24 h at room temperature (RT). At the end of the treatments, they were rinsed with double distilled water and the seeds were air dried at room temperature up to the original moisture content (48 h). Primed and not-primed seeds were stored at +4 °C until the experiments.

Before sowing, the surface sterilized seeds were germinated in Petri dishes (10 seeds each) containing two layers of filter paper soaked with 15 mL of double distilled water. The Petri dishes were incubated in the dark for 7 days at RT. Then, the germinated seeds were carefully sown in plastic pots (15 cm diameter), 5 seeds per pot containing about 300 g of soil (COMPO SANA^®^ COMPACT, Germany. Soil characteristics: pH 6.5; dry bulk density 150 kg/m^3^; electrical conductivity: 0.50 dS/m; porosity 90% *v*/*v*. Soil components: neutral sphagnum peat, perlite (<5%), composted green soil improver). The plants (5 seedlings/pots) were grown in natural sunlight (light intensity: 95.9 mmol/day ± 6.7 mmol/day), at temperature of 25 °C ± 0.4 °C and soil moisture of 54.7% ± 3.1%.

Seedlings were grown for 14 days before the beginning of salt treatments. Saline water consisted of the following solutions: 80 mM NaCl (EC: 8.78 dS/m) and 160 mM NaCl (EC: 16.23 dS/m) The pots were randomly assigned to the experimental sets: (1) non-primed seeds irrigated with tap water (EC: 0.62 dS/m) (control) or with saline solutions; (2) primed seeds irrigated either with tap water (EC: 0.62 dS/m) or with saline solutions. Primed and non-primed plants were watered with 80 mL of tap water or salt solution every 72 h for 4 weeks. The pots were randomly moved at the time of watering.

### 4.3. Soil Analysis

The chemical-physical parameters of the soils were determined in pots containing plants and in pots without plants (blank) in order to evaluate the absorption of minerals by the roots.

The gravimetric water content of soil was determined as reported by Santangeli et al. [28] at the end of the experiments. The percentage of water content was determined by oven-drying soils samples at a temperature of 70 °C for 48 h. The average (%) of water was estimated as:Gravimetric Water Content (%) = ((f.w. − d.w.)/d.w.) × 100
where: f.w. = Soil fresh weight; d.w. = Soil dry weight.

The determination of the electrical conductivity (EC) was carried out according to Sairam et al. [56]. Soil samples (500 mg dry weight) were dipped in water milli-Q (0.1 mL H_2_O mg d.w.^−1^). They were subsequently incubated in a water bath at 100 °C for 15 min and then filtered with paper funnels. Electrical conductivity (EC) was measured on the filtrate at room temperature with an EC meter (HANNA Instrument 98312 DiST^®^5 & DiST^®^6, Padova, Italy).

### 4.4. Morphological Parameters, Tolerance Index and Plant Water Content

Morphological parameters were monitored at the end of the saline treatments to evaluate the effects of salinity on plant growth. Parameters taken in account were length of the stem (cm), number of leaves and length of the longest root (cm).

After 42 days of growth, plants were harvested and tolerance index (TI), root and shoot toxicity were calculated according to Idrees et al. [57], using the following formula:TI (%) = (Mean root length in stress/Mean root length in control) × 100

Plant water content was determined according to Zeng et al. [58]. Plants were dried at 70 °C for 48 h. The average (%) of water content was calculated according to the following formula:Water Content (%) = ((f.w. − d.w.)/f.w.) × 100
where f.w. = Plant fresh weight; d.w = Plant dry weight

Plants were sampled (200 mg fresh weight, unless differently reported) and were frozen by dipping in liquid nitrogen and stored at −20 °C until further analyses.

### 4.5. Chlorophylls

Frozen samples were homogenized in liquid nitrogen and resuspended in 95% ethanol. Samples were incubated in the dark, to avoid chlorophyll degradation, overnight at 4 °C, then centrifuged at 8000× *g* for 15 min. The supernatants were collected and stored at −20 °C until the analysis. To quantify chlorophyll content, the absorbances of the supernatants were determined by a spectrophotometer (VARIAN Cary 50 Bio, Santa Clara, CA, USA) at 664.1 nm (chlorophyll a) and 648.6 nm (chlorophyll b). The concentration of photosynthetic pigments was determined according to Lichtenthaler [59]. The amount of pigment was expressed as μg ⋅ g f.w.^−1^.

### 4.6. Soluble Sugars

Quantification of monosaccharides was performed according to the Anthrone protocol by Chun and Yin [60] with major modifications. Frozen samples were homogenized in liquid nitrogen and resuspended in 1 mL of 1% phosphate saline buffer (PBS). Samples were incubated in the dark overnight at 4 °C, then centrifuged at 6200× *g* for 20 min, after which 0.5 mL of each supernatant was collected and stored at −20 °C until the analysis.

Next, 0.5 mL of 30% KOH was added to the homogenate in a vial and kept at 100 °C in a water bath for 35 min. After the addition of 1.5 mL of 95% ethanol, the extracts were centrifuged at 4000× *g* for 15 min and the supernatants were discarded. The pellets were resuspended in 0.5 mL of distilled water and transferred to a glass tube. Reaction started by adding 2.5 mL of fresh solution of 0.2% anthrone in 75% H_2_SO_4_ to standard solutions and samples. Glass tubes were put on ice for few seconds, placed in a boiling water bath for 10 min and back in ice. The absorbance of the reaction mixture was measured at 625 nm with a spectrophotometer. The concentration of monosaccharides was calculated according to a calibration curve of glucose, carried out with solutions of 20 mg L^−1^, 40 mg L^−1^, 60 mg L^−1^, 80 mg L^−1^ and 100 mg L^−1^ (y = 0.0063x + 0.0236; R^2^ = 0.9955). The sugar content was calculated as mg glucose equivalent g f.w.^−1^.

### 4.7. Secondary Metabolites and Proline Content

Sample preparation for phenols, flavonoids and proline quantification was performed by grinding the frozen material in liquid nitrogen and by resuspending in 5 mL of 95% ethanol. Samples were incubated in the dark overnight at 4 °C, then centrifuged at 8000× *g* for 15 min. The supernatants were collected and stored at −20 °C until the analysis.

#### 4.7.1. Phenolic Compounds

The total phenolic content was determined according to Santangeli et al. [28], using Folin–Ciocalteau reagent—50 µL of each standard solutions and samples were separately mixed with 0.475 mL of 0.25 N Folin, incubated in dark for 2 min and then 0.475 mL of 1 M sodium carbonate were added. After 1 h of incubation in the dark at room temperature, the absorbance of the reaction mixture was measured at 724 nm with a spectrophotometer. The concentration of phenolic compounds was calculated according to a calibration curve of chlorogenic acid carried out with solutions of 10 μg mL^−1^, 20 μg mL^−1^, 40 μg mL^−1^, and 50 μg mL^−1^ (y = 0.006x − 0.0349; R^2^ = 0.996). The total phenolic content was calculated as μg chlorogenic acid equivalent ⋅ g f.w.^−1.^

#### 4.7.2. Flavonoids

Flavonoids were estimated according to Chang et al. [61] with slight modifications—0.5 mL of each standard solutions and samples were separately mixed with 1.5 mL of reaction solution (95% ethanol, 0.1 mL of 10% aluminum chloride, 0.1 mL of 1 M potassium acetate, 2.8 mL of distilled water). After incubation at RT for 30 min, the absorbance was measured at 415 nm with a spectrophotometer. To calculate the concentration of flavonoids, a calibration curve was performed using different concentrations of quercetin as standard (10 μg mL^−1^, 20 μg mL^−1^, 40 μg mL^−1^, and 80 μg mL^−1^; y = 0.0081x − 0.0321; R^2^ = 0.999). Flavonoids are expressed as μg of quercetin equivalent ⋅ g f.w.^−1^.

#### 4.7.3. Proline

The quantitative determination of proline was performed according to the protocol of Stassinos et al. [26]—500 μL of the extract was added to 1 mL of reaction mixture composed by ninhydrin 1% (*w*/*v*) dissolved in a mixture of 60% (*v*/*v*) acetic acid and 20% (*v*/*v*) ethanol. The samples, protected from light, were heated at 95 °C for 20 min. Proline concentration was evaluated by detecting the absorbance at 520 nm with a spectrophotometer. To calculate the osmolyte concentration, a calibration curve was performed using standard solutions of L-Proline from 5, 10, 15, and 20 μg mL^−1^ (y = 0.0684x − 0.0624; R^2^ = 0.996). Data are expressed as μg proline ⋅ g f.w.^−1^.

### 4.8. Thiobarbituric acid Reactive Products

Thio-barbituric acid reactive products (TBA) were estimated according to Micheli et al. [62] and Kaur and Jindal [63] with major modifications. A fresh solution of 0.5% (*w*/*v*) TBA in 20% (*w*/*v*) of trichloroacetic acid (TCA) was prepared and kept in the dark, just before the assay. Frozen material (100 mg of fresh weight) was homogenized in 0.7 mL of ddH_2_O, then 0.75 mL of 0.5% TBA were added, and the samples were incubated at 50 °C for 50 min. Blank was prepared by adding in a test tube ddH_2_O and TBA, without sample. The reaction was stopped by placing the samples in ice for 10 min. The sample was transferred to a glass cuvette and the absorbances (at 532 nm and 600 nm), were detected, using the spectrophotometer. TBA reactive species were calculated as malondialdehyde (MDA) equivalent, according to the following formula:MDA equivalent (mmol/L) = ((Abs_532_ − Abs_600_)/(ε × l))
where ε = extinction coefficient of MDA at 532 nm (155 mM^−1^ cm^−1^); l = path length of cuvette (1 cm). Data were expressed as mmol MDA equivalent ⋅ g f.w.^−1^

### 4.9. Antioxidant Activity

Antioxidant and reducing power and scavenger activity were measured with the DPPH, PFRAP and FRAP assay, respectively.

#### 4.9.1. 2,2-Diphenyl-1-picryl-hydrazyl-hydrate (DPPH) Free Radical Assay

The percentage of antioxidant activity of each sample was assessed by DPPH free radical assay. The measurement of the DPPH radical scavenging activity was performed according to the protocol described by Garcia et al. [64]. Frozen samples were homogenized in liquid nitrogen and resuspended in 5 mL of 95% ethanol. Samples were incubated in the dark overnight at 4 °C, then centrifuged at 8000× *g* for 15 min and treated as reported by Garcia et al. [64].

#### 4.9.2. Potassium Ferricyanide and Ferric Reducing Antioxidant Power (PFRAP and FRAP) Assays

Sample preparation for FRAP and PFRAP assay required the same methodology: frozen samples were homogenized in 1.5 mL of methanol. Samples were incubated at room temperature overnight, then centrifuged at 4000× *g* for 15 min. The supernatants were collected and stored at 4 °C until the analysis.

PFRAP assay was based on the procedure of Hue et al. [65] with modifications—50 μL of extracts were added to 200 μL of 1% PBS and 200 μL of 1% (*w*/*v*) potassium ferricyanide. The mixture was incubated for 20 min at 50 °C and trichloroacetic acid (250 μL) was added to the mixture. The supernatant (500 μL) was added to 500 μL of deionized water and 100 μL of 0.1% (*w*/*v*) ferric chloride. The mixture was incubated at 37 °C for 10 min and the absorbance was recorded at 700 nm by spectrophotometer. Ascorbic acid was used as positive control. The scavenging activity was expressed as the %, compared to ascorbic acid, set at 100% activity.

FRAP assay was based on the procedures of Gohari et al. [66] and Lim and Lim [67], with modifications. The FRAP reagent was prepared as described by [66], mixing 45 mL of 0.3 M acetate buffer pH 3.6, 4.5 mL of 10 mM TPTZ solution solubilized in 40 mM HCl, and 4.5 mL of 20 mM FeCl_3_ solution, in a 10:1:1 ratio. FRAP solution (3.6 mL) was added to distilled water (0.4 mL) and incubated at 37 °C for 5 min. The solution was mixed with 0.2 mL of plant extract and incubated at 37 °C for 10 min. The absorbance was measured at 593 nm with the spectrophotometer. The ferric reducing power was calculated according to a calibration curve, made with known concentrations of FeSO_4_·7H_2_O (0.1, 0.4, 0.8, 1, 1.5 mM) (y = 1.0654x − 0.0263; R^2^ = 0.9966) and calculated as mmol FeSO_4_ equivalent g f.w.^−1^

### 4.10. Enzymatic Activities

Enzyme activities were related to the amount of protein in the sample examined. The protein concentration was determined by Bradford [68] and calculated using a calibration curve with bovine serum albumin (BSA) (1.25; 2.5; 5; 7.5 and 10 μg mL^−1^) (y = 0.0319x + 0.0285; R^2^ = 0.9922).

#### 4.10.1. Peroxidase 

Peroxidase (POD) (EC 1.11.1.7) activity was analyzed as described by Yang et al. [69] with minor modifications. The activity was determined by the pyrogallol oxidation method in the presence of H_2_O_2_. A unit of enzyme activity was defined as the amount of oxidation of pyrogallol into purpurogallin at pH 6 in 20 s.

Frozen samples were homogenized in liquid nitrogen and resuspended in 1 mL of 1% PBS. After 1 h in ice, the homogenates were centrifuged at 6200× *g*, and the supernatants were used to detect peroxidase activity. 0.32 mL pyrogallic acid (5%, *w*/*v*), 0.16 mL H_2_O_2_ (0.5%, *v*/*v*), 0.32 mL potassium sulfate buffer (0.1 mol L^−1^, pH 6.0) and 2.2 mL distilled water were added. The absorbance at 420 nm was recorded as the blank control, then 0.05 mL of supernatant was added into the cuvette, mixed and recorded the absorbance at 420 nm for 3 min, using the spectrophotometer.

The enzyme activity was calculated according to Yang et al. [69] and expressed as enzymatic unit (U mg protein^−1^).

#### 4.10.2. Polyphenol Oxidase 

Polyphenol oxidase (PPO) (EC 1.14.18.1) activity was determined according to the method of Orzali et al. [70], with minor modifications. Frozen samples were homogenized in 2 mL of 50 mM sodium-phosphate buffer (pH 6.5) containing 1% (*w*/*v*) polyvinylpoli-pyrrolidone (PVPP), 2 mM Na EDTA, 6% (*v*/*v*) Triton X-100, 0.02% (*w*/*v*) MgCl_2_ and proteases inhibitor cocktail (Sigma). The homogenate was centrifuged at 5000× *g* for 15 min at 4 °C. 0.1 mL of sample was added to 1.4 mL reaction mixture (50 mM sodium-phosphate buffer, pH 6.5) containing 50 mM pyrocatechol as substrate. The kinetics were followed by spectrophotometer at the wavelength of 420 nm for 300 s. The enzymatic activity of PPO was expressed as enzymatic unit (U mg protein^−1^).

#### 4.10.3. Superoxide Dismutase

The enzymatic activities were determined as described by Santangeli et al. [28]. Frozen homogenized samples, with polyvinylpolypyrrolidone (PVPP), were suspended in 1% PBS with protease inhibitor cocktail for plant cells (Sigma), incubated overnight at 4 °C, centrifuged at 10,000× *g* at 4 °C for 30 min and the supernatants were recovered and stored at −20 °C until analysis. Superoxide dismutase (SOD) (EC 1.15.1.1) activity was assayed by NPAGE (native polyacrylamide gel electrophoresis). Samples (40 μg of proteins) were loaded and separated on native polyacrylamide gels. SOD activity was visualized following the procedure described by Beauchamp and Fridovich [71]. SOD activity was expressed as Arbitrary Units (A.U.), which corresponds to the pixel density of each lane obtained by the program Image J 1.53A.

#### 4.10.4. Ascorbate Peroxidase

Ascorbate peroxidase (APX) (EC 1.11.1.11) activity was determined according to the method of Orzali et al. [70]. Frozen samples were ground in the extraction buffer (1 M sodium-phosphate buffer pH 7, 1% (*w*/*v*) PVPP, 3 mM EDTA, 0.5 mM ascorbic acid, 0.1% Triton X-100 and proteases inhibitor cocktail). The homogenate was centrifuged at 5000× *g* for 20 min at 4 °C. The reaction was induced by the addition of 1 mM H_2_O_2_ in 1 mM EDTA and 0.5 mM of ascorbic acid as substrate. The rate of ascorbate oxidation was followed for 150 s at 290 nm by spectrophotometer. Enzyme activity was expressed as the %, compared to the untreated control (0 mM NaCl).

### 4.11. Quantification of Intracellular Free Calcium and Determination of Transglutaminase (TGase) Enzyme Activity

Samples were homogenized using liquid nitrogen and resuspended in 1.5 mL of tricine buffer (0.2 M pH 8). The extracts were incubated overnight at 4 °C, then centrifuged at 12,000× *g* at 4 °C for 10 min and the supernatants were recovered and filtered (0.45 μm pore size, Millipore) and stored at −20 °C until analysis.

The quantification of calcium was performed using the Calcium Assay Colorimetric Kit (Abcam, Cambridge, UK—ab272527; www.abcam.com/ab272527 (accessed on 15 December 2022)) and a microplate reader (Spark^®^ Multimode Microplate Reader—Tecan, Switzerland). Data are expressed as μg Ca^2+^ ⋅ mg f.w.^−1^.

The TGase activity was monitored, using the Fluorescent Transglutaminase Assay Kit (T036—Zedira GmbH, Darmstadt, Germany) and a microplate reader (Spark^®^ Multimode Microplate Reader—Tecan, Switzerland) by measuring the fluorescence (excitation wavelength 332 nm; emission wavelength 500 nm). The relative TGase activity was detected by the increasing of fluorescence intensity over time. Enzyme activity is expressed as the %, compared to the untreated control.

### 4.12. Statistical Analysis

Data are expressed as mean ± standard error (SE). One-way analysis of variance (ANOVA) was performed with Past 4.11. The Tukey–Kramer method was used to assess the difference of significance among groups. All analyses were considered significant at *p* < 0.05 within each treatment group. When comparing primed groups to non-primed, the significance was *** *p* < 0.001; ** *p* < 0.01; * *p* < 0.05.

## 5. Conclusions

Seed priming can be a tool to increase tomato salt tolerance, leading to a better growth, especially during the first weeks of the life cycle. Tomato seed priming with PAs (PUT, SPD and SPM) results in improved plant tolerance to salt stress. Polyamines have the advantage of being naturally occurring and non-toxic compounds, encouraging their application in increasing crop yield and quality without any negative effect to crops and/or the environment. In our work, we provide evidence that tomato acclimation to irrigation with saline solution is improved by seed treatment with PAs. The use of PAs priming may find future practical applications for the protection of tomato crop from different stresses.

## Figures and Tables

**Figure 1 plants-12-01855-f001:**
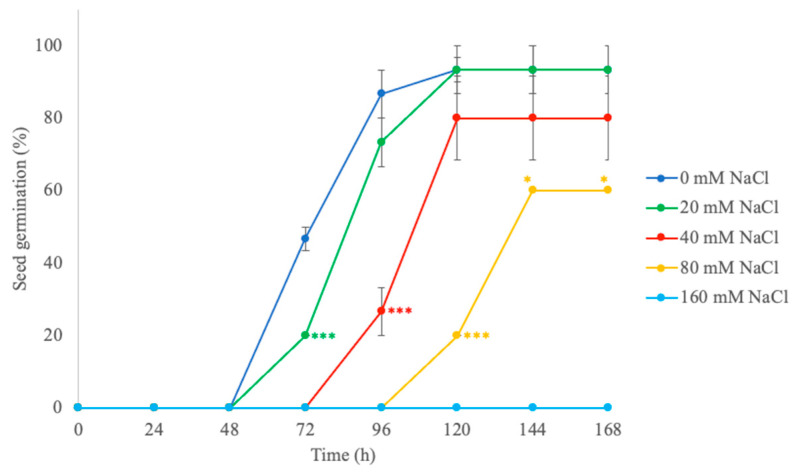
Effect of different saline solutions (0 mM NaCl, 20 mM NaCl, 40 mM NaCl, 80 mM NaCl, 160 mM NaCl) on seed germination (%) of tomato cv. Principe Borghese. Data are expressed as means ± SE (*n* = 6). Significant differences to the control (CTRL; 0 mM NaCl) are reported as * *p* < 0.05; *** *p* < 0.001.

**Figure 2 plants-12-01855-f002:**
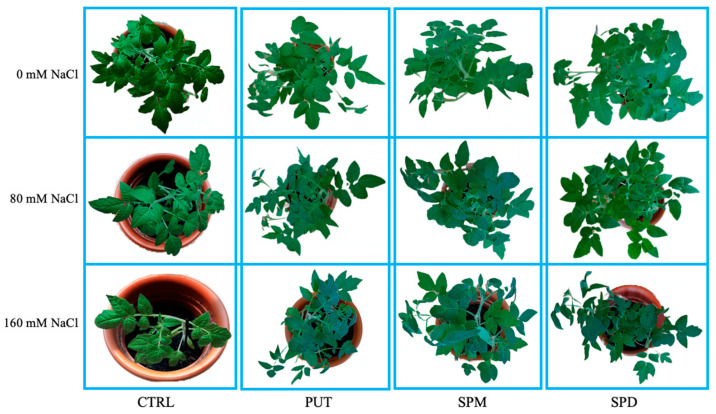
Tomato plants irrigated with different saline solutions (0 mM NaCl, 80 mM NaCl, 160 mM NaCl), at the end of the experiments. PUT = putrescine; SPM = spermine; SPD = spermidine.

**Figure 3 plants-12-01855-f003:**
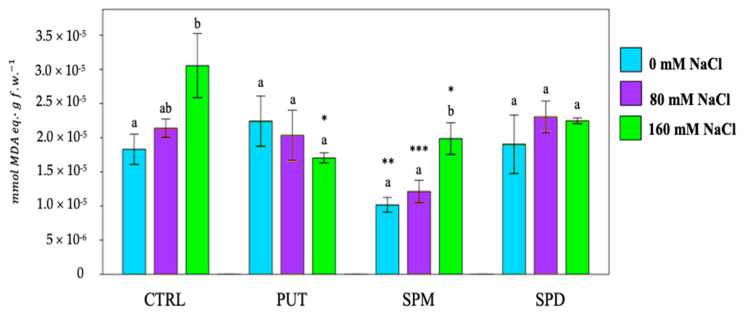
Thio-barbituric acid reactive products of tomato plants. PUT = putrescine; SPM = spermine; SPD = spermidine. Data are expressed as mean ± SE (*n* = 9). Mean values in the column marked by different letters are significantly different within the same group (*p* < 0.05; ANOVA and Tukey–Kramer test). Significant differences to CTRL are reported as * *p* < 0.05; ** *p* < 0.01; *** *p* < 0.001.

**Figure 4 plants-12-01855-f004:**
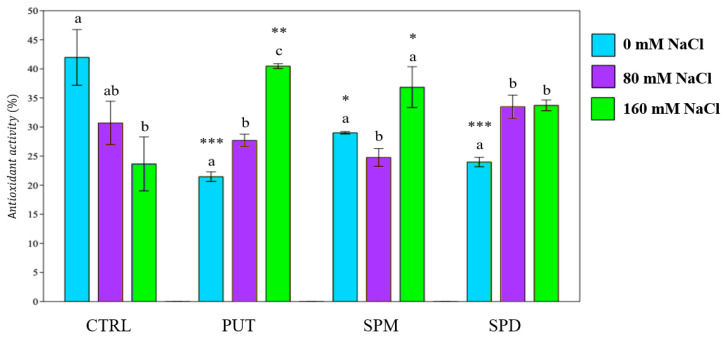
Antioxidant activity of tomato plants. PUT = putrescine; SPM = spermine; SPD = spermidine. Data are expressed as mean ± SE (*n* = 6). Mean values in the column marked by different letters are significantly different within the same group (*p* < 0.05; ANOVA and Tukey–Kramer test). Significant differences to CTRL are reported as * *p* < 0.05; ** *p* < 0.01; *** *p* < 0.001.

**Figure 5 plants-12-01855-f005:**
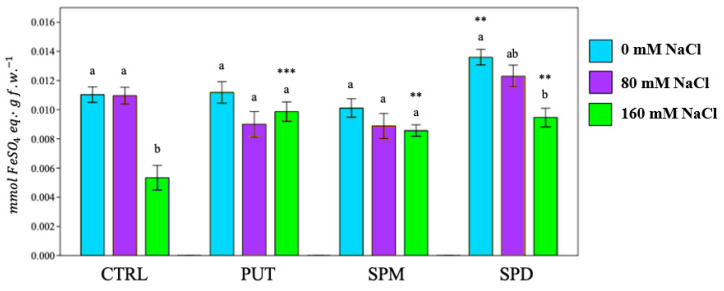
Ferric reducing antioxidant power of tomato plants. PUT = putrescine; SPM = spermine; SPD = spermidine. Data are expressed as mean ± SE (*n* = 6). Mean values in the column marked by different letters are significantly different within the same group (*p* < 0.05; ANOVA and Tukey–Kramer test). Significant differences to CTRL are reported as ** *p* < 0.01; *** *p* < 0.001.

**Figure 6 plants-12-01855-f006:**
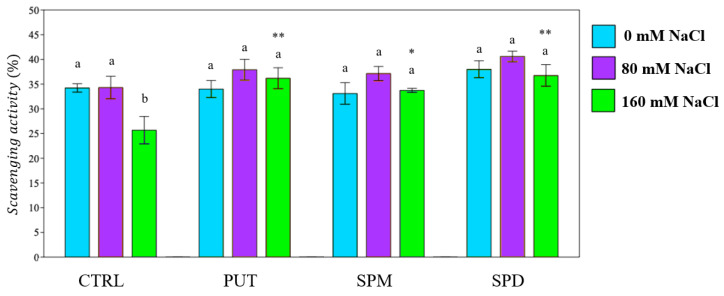
Potassium ferricyanide reducing antioxidant power of tomato plants, exposed to different saline solutions. PUT = putrescine; SPM = spermine; SPD = spermidine. Data are expressed as mean ± SE (*n* = 6). Mean values in the column marked by different letters are significantly different within the same group (*p* < 0.05; ANOVA and Tukey–Kramer test). Significant differences to CTRL are reported as * *p* < 0.05; ** *p* < 0.01. The activity was expressed as the %, compared to ascorbic acid (set at 100% activity).

**Figure 7 plants-12-01855-f007:**
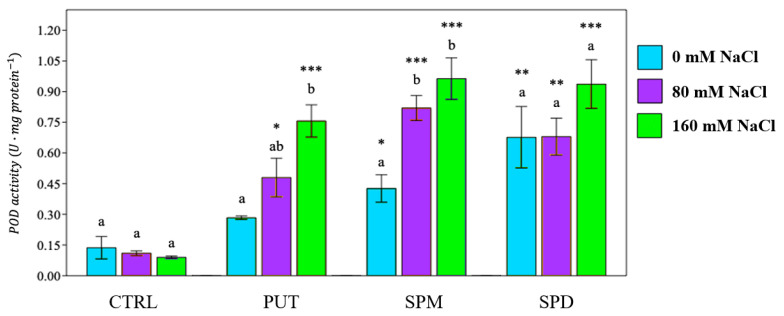
POD activity of primed and non-primed tomato plants. PUT = putrescine; SPM = spermine; SPD = spermidine. Data are expressed as mean ± SE (*n* = 6). Mean values in the column marked by different letters are significantly different within the same group (*p* < 0.05; ANOVA and Tukey–Kramer test). Significant differences to CTRL are reported as * *p* < 0.05; ** *p* < 0.01; *** *p* < 0.001.

**Figure 8 plants-12-01855-f008:**
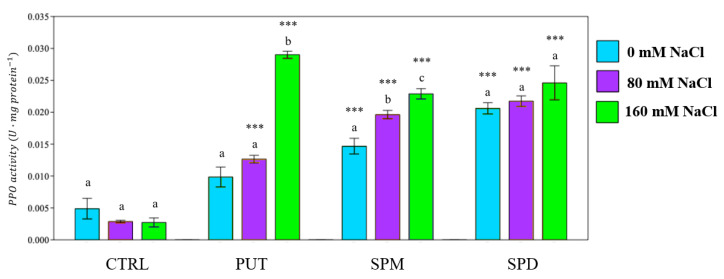
PPO activity of tomato plants exposed to different saline solutions. PUT = putrescine; SPM = spermine; SPD = spermidine. Data are expressed as mean ± SE (*n* = 6). Mean values in the column marked by different letters are significantly different within the same group (*p* < 0.05; ANOVA and Tukey–Kramer test). Significant differences to CTRL are reported as *** *p* < 0.001.

**Figure 9 plants-12-01855-f009:**
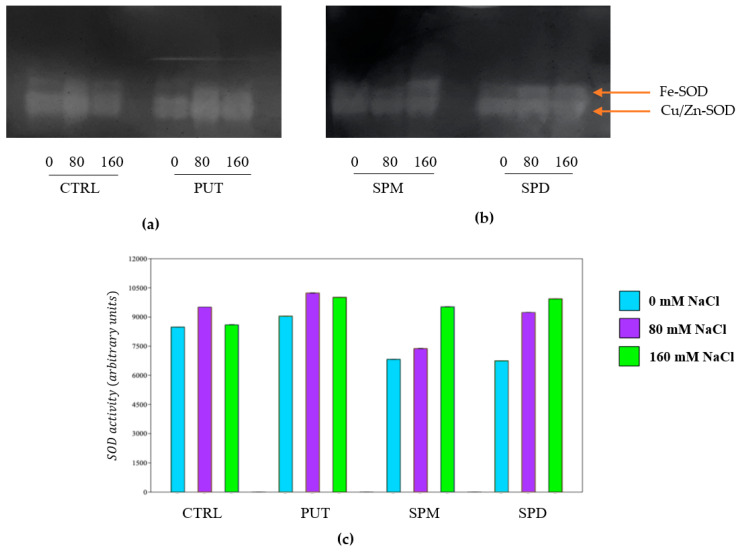
Native gels of tomato plant extracts: (**a**) controls (CTRL) and primed with PUT; (**b**) SPM and SPD; (**c**) SOD activity of tomato plants, expressed as Arbitrary Units.

**Figure 10 plants-12-01855-f010:**
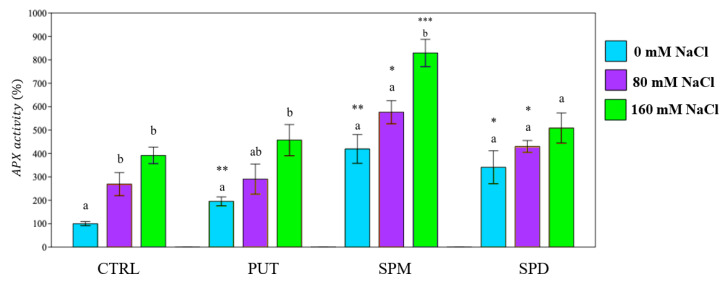
APX activity of tomato plants under different saline conditions. PUT = putrescine; SPM = spermine; SPD = spermidine. Data are expressed as mean ± SE (*n* = 6). Mean values in the column marked by different letters are significantly different within the same group (*p* < 0.05; ANOVA and Tukey–Kramer test). Significant differences to CTRL are reported as * *p* < 0.05; ** *p* < 0.01; *** *p* < 0.001.

**Figure 11 plants-12-01855-f011:**
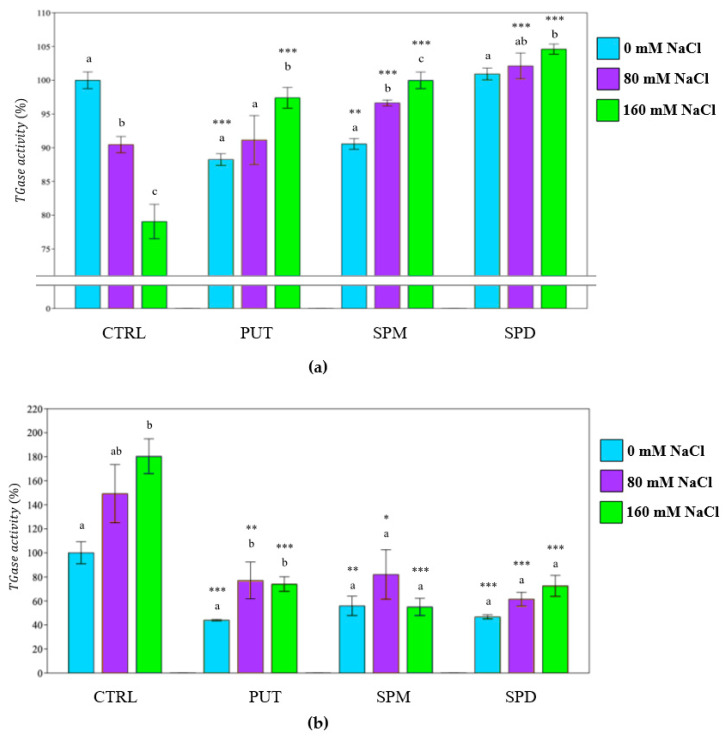
TGase activity in (**a**) shoots and (**b**) roots of tomato plants. PUT = putrescine; SPM = spermine; SPD = spermidine. Data are expressed as mean ± SE (*n* = 9). Mean values in the column marked by different letters are significantly different within the same group (*p* < 0.05; ANOVA and Tukey–Kramer test). Significant differences to CTRL are reported as * *p* < 0.05; ** *p* < 0.01; *** *p* < 0.001.

**Table 1 plants-12-01855-t001:** Effect of different priming solutions on seed germination (%). Data are expressed as mean ± SE (*n* = 10). Mean values in the column, marked by different letters, are significantly different within the same group (*p* < 0.05; ANOVA and Tukey–Kramer test).

Priming Solutions	48 h	72 h	96 h	120 h
CTRL	21.1 ± 4.1 ^a^	53.7 ± 6.9 ^b^	83.8 ± 4.3 ^a^	91.3 ± 2.2 ^a^
2.5 mM PUT	7 ± 3.7 ^a^	80.3 ± 2.9 ^a^	84.9 ± 3.6 ^a^	87.7 ± 3.7 ^a^
2.5 mM SPM	12 ± 6.1 ^a^	83.5 ± 4.4 ^a^	88.5 ± 4 ^a^	91.2 ± 3.7 ^a^
2.5 mM SPD	16 ± 8.2 ^a^	84.9 ± 4.4 ^a^	88.8 ± 4.2 ^a^	89.7 ± 4.1 ^a^

**Table 2 plants-12-01855-t002:** Effect of different priming solutions on seed germination (%), one year after the treatments. Data are expressed as mean ± SE (*n* = 3). Mean values in the column marked by different letters are significantly different within the same group (*p* < 0.05; ANOVA and Tukey–Kramer test).

Priming Solutions	72 h	96 h	120 h	144 h	168 h
CTRL	0 ^a^	0 ^a^	0 ^a^	0 ^a^	19.4 ± 7.1 ^a^
2.5 mM PUT	18.3 ± 1.7 ^b^	45 ± 2.9 ^b^	77.8 ± 11.1 ^b^	88.9 ± 11.1 ^b^	88.9 ± 11.1 ^b^
2.5 mM SPM	0 ^a^	31.7 ± 1.7 ^c^	38.9 ± 5.6 ^c^	44.4 ± 5.6 ^c^	56.7 ± 3.3 ^c^
2.5 mM SPD	15.7 ± 0.7 ^b^	52.3 ± 1.5 ^b^	73.3 ± 8.3 ^b^	73.3 ± 8.3 ^b^	73.3 ± 8.3 ^bc^

**Table 3 plants-12-01855-t003:** Gravimetric Water Content (GWC) and Electrical Conductivity (EC) of the soils at the end of the experiments. Data are expressed as mean ± SE (*n* = 6). Mean values in the column marked by different letters are significantly different within the same group (*p* < 0.05; ANOVA and Tukey–Kramer test). Significant differences to control (CTRL) are reported as * *p* < 0.05; ** *p* < 0.01; *** *p* < 0.001.

Priming Solutions	NaCl (mM)	GWC (%)	EC (dS/m)
CTRL	0	40.93 ± 4.37 ^a^	0.48 ± 0.04 ^a^
	80	45.65 ± 1.22 ^a^	1.05 ± 0.05 ^b^
	160	46.90 ± 4.61 ^a^	1.84 ± 0.08 ^c^
2.5 mM PUT	0	25.55 ± 5.95 ^a^ *	0.37 ± 0.02 ^a^ *
	80	27.64 ± 5.09 ^a^ **	1.18 ± 0.13 ^b^
	160	46.96 ± 3.67 ^b^	2.30 ± 0.18 ^c^ *
2.5 mM SPM	0	17.05 ± 3.50 ^a^ **	0.37 ± 0.04 ^a^ *
	80	34.45 ± 3.63 ^b^ *	1.00 ± 0.07 ^b^
	160	50.99 ± 4.94 ^c^	2.17 ± 0.12 ^c^ *
2.5 mM SPD	0	20.31 ± 3.35 ^a^ **	0.37 ± 0.02 ^a^ *
	80	36.23 ± 5.18 ^b^ *	1.20 ± 0.11 ^b^
	160	46.22 ± 5.59 ^c^	1.22 ± 0.08 ^b^ ***

**Table 4 plants-12-01855-t004:** Morphological parameters (number of leaves, shoot and root length), stress Tolerance Index (TI), Plant Water Content (PWC) and Biomass of tomato, grown under different saline conditions. Data are expressed as mean ± SE (*n* = 12) except for PWC (*n* = 8). Mean values in the column marked by different letters are significantly different within the same group (*p* < 0.05; ANOVA and Tukey–Kramer test). Significant differences to CTRL are reported as * *p* < 0.05; ** *p* < 0.01; *** *p* < 0.001.

Priming Solutions	NaCl (mM)	N. Leaves	Shoot Lenght (cm)	Root Lenght (cm)	TI (%)	PWC (%)	Biomass (g)
CTRL	0	37 ± 4 ^a^	35.2 ± 3.3 ^a^	11.7 ± 0.9 ^a^	100	95.4 ± 0.6 ^a^	5.2 ± 0.9 ^a^
	80	27 ± 2 ^ab^	24.2 ± 1.3 ^b^	9.6 ± 0.8 ^ab^	82 (−18%)	94 ± 1.1 ^a^	3.2 ± 0.4 ^ab^
	160	21 ± 2 ^b^	15.4 ± 1.6 ^b^	5.3 ± 0.6 ^b^	45 (−55%)	91.6 ± 0.8 ^b^	2.1 ± 0.3 ^b^
2.5 mM PUT	0	49 ± 1 ^a^ *	44.3 ± 0.5 ^a^ *	14.1 ± 1.2 ^a^	121 (+21%)	93.7 ± 0.7 ^a^ *	2.9 ± 1.4 ^a^
	80	42 ± 3 ^ab^ **	36.3 ± 1.9 ^a^ **	14.4 ± 1.1 ^a^ *	123 (+41%)	93.6 ± 0.7 ^a^	4.6 ± 1.5 ^a^
	160	31 ± 2 ^b^ *	25.6 ± 0.9 ^b^ *	14.5 ± 1.2 ^a^ ***	124 (+79%)	93.6 ± 1 ^a^	4 ± 1.1 ^a^
2.5 mM SPM	0	42 ± 1 ^a^	45 ± 2.9 ^a^ *	13 ± 1.8 ^a^	111 (+11%)	92.8 ± 0.5 ^a^ *	5.4 ± 1.3 ^a^
	80	41 ± 1 ^a^ **	38.5 ± 1.7 ^ab^ ***	12.6 ± 1.4 ^a^	108 (+26%)	94.6 ± 0.9 ^a^	3.6 ± 1.4 ^a^
	160	28 ± 2 ^b^	34 ± 0.4 ^b^ ***	14.9 ± 1.4 ^a^ ***	127 (+82%)	91.4 ± 2.6 ^a^	2.6 ± 0.7 ^a^
2.5 mM SPD	0	44 ± 1 ^a^	46.3 ± 1.6 ^a^ **	13 ± 0.7 ^a^	111 (+11%)	90.4 ± 0.9 ^a^ **	6.1 ± 1.7 ^a^
	80	39 ± 2 ^ab^ *	38.8 ± 1.2 ^ab^ ***	12.5 ± 1.3 ^a^	107 (+25%)	93.5 ± 1.3 ^a^	5.7 ± 2 ^a^
	160	29 ± 2 ^b^ *	31.5 ± 0.9 ^b^ ***	12.5 ± 1 ^a^ **	107 (+62%)	90.6 ± 1.1 ^a^	2.6 ± 0.8 ^a^

**Table 5 plants-12-01855-t005:** Chlorophyll *a* (Chl a), Chlorophyll *b* (Chl b) and Total Chlorophyll content (Total Chl) of tomato, exposed to different saline conditions. Data are expressed as means ± SE (*n* = 6). Mean values in the column marked by different letters are significantly different within the same group (*p* < 0.05; ANOVA and Tukey–Kramer test). Significant differences between groups are reported as * *p* < 0.05; ** *p* < 0.01; *** *p* < 0.001.

Priming Solution	NaCl (mM)	Chl a (μg/g f.w.)	Chl b (μg/g f.w.)	Total Chl (μg/g f.w.)
CTRL	0	88.60 ± 2.24 ^a^	48.40 ± 5.64 ^a^	137 ± 4.90 ^a^
	80	89.69 ± 6.86 ^a^	34.94 ± 5.41 ^a^	124.63 ± 7.72 ^ab^
	160	80.86 ± 4.12 ^a^	34.63 ± 3.35 ^a^	115.49 ± 5.09 ^b^
2.5 mM PUT	0	98.07 ± 7.21 ^a^	37.03 ± 4.44 ^a^	135.10 ± 3.19 ^a^
	80	118.76 ± 5.19 ^ab^ **	47.16 ± 5.27 ^ab^ *	166.37 ± 3.31 ^b^ ***
	160	126.22 ± 1.73 ^b^ ***	50.79 ± 0.50 ^b^ ***	177.01 ± 1.50 ^c^ ***
2.5 mM SPM	0	78.08 ± 5.62 ^a^	33.88 ± 5.65 ^a^	111.96 ± 3.31 ^a^ *
	80	126.82 ± 7.20 ^b^ ***	56.06 ± 7.72 ^b^ *	182.88 ± 4.04 ^b^ ***
	160	143.25 ± 4.87 ^b^ ***	58.57 ± 2.30 ^b^ ***	201.82 ± 3.32 ^c^ ***
2.5 mM SPD	0	112.39 ± 5.77 ^a^ *	51.46 ± 7.57 ^a^	163.85 ± 2.67 ^a^ **
	80	123.52 ± 2.42 ^b^ ***	51.97 ± 4.43 ^a^ *	175.49 ± 5.87 ^b^ ***
	160	123.09 ± 2.90 ^b^ ***	63.23 ± 4.36 ^a^ ***	186.31 ± 6.49 ^c^ ***

**Table 6 plants-12-01855-t006:** Soluble sugars and Proline content of tomato plants, exposed to different saline solutions. PUT = putrescine; SPM = spermine; SPD = spermidine. Data are expressed as mean ± SE (*n* = 6). Mean values in the column marked by different letters are significantly different within the same group (*p* < 0.05; ANOVA and Tukey–Kramer test). Significant differences to CTRL are reported as * *p* < 0.05; ** *p* < 0.01; *** *p* < 0.001.

Priming Solutions	NaCl (mM)	Soluble Sugars (mg Glucose eq. g f.w.^−1^)	Proline (μg Proline eq. g f.w.^−1^)
CTRL	0	0.269 ± 0.057 ^a^	320.6 ± 12.4 ^a^
	80	1.244 ± 0.047 ^b^	356.3 ± 21.2 ^a^
	160	1.458 ± 0.034 ^c^	369.2 ± 10.4 ^a^
2.5 mM PUT	0	0.330 ± 0.022 ^a^	126.8 ± 6.5 ^a^ ***
	80	1.133 ± 0.073 ^b^	224 ± 6 ^b^ ***
	160	1.422 ± 0.024 ^c^	802.6 ± 29.2 ^c^ ***
2.5 mM SPM	0	0.199 ± 0.005 ^a^	131 ± 7 ^a^ ***
	80	1.022 ± 0.043 ^b^ *	242.4 ± 12.62 ^b^ ***
	160	1.159 ± 0.066 ^b^ ***	652.6 ± 30.13 ^c^ ***
2.5 mM SPD	0	0.270 ± 0.015 ^a^	112.6 ± 5.8 ^a^ ***
	80	0.985 ± 0.025 ^b^ **	520.3 ± 19.6 ^b^ ***
	160	1.189 ± 0.052 ^b^ **	620.4 ± 5.3 ^c^ ***

**Table 7 plants-12-01855-t007:** Phenolic compounds and Flavonoids of tomato plants. PUT = putrescine; SPM = spermine; SPD = spermidine. Data are expressed as mean ± SE (*n* = 6). Mean values in the column marked by different letters are significantly different within the same group (*p* < 0.05; ANOVA and Tukey–Kramer test). Significant differences to CTRL are reported as * *p* < 0.05; ** *p* < 0.01; *** *p* < 0.001.

Priming Solutions	NaCl (mM)	Phenols (μg Chlorogenic Acid eq. g f.w.^−1^)	Flavonoids (μg Quercetin eq. g f.w.^−1^)
CTRL	0	866.61 ± 22.83 ^a^	2.24 ± 0.06 ^a^
	80	721.74 ± 23.72 ^b^	1.82 ± 0.06 ^b^
	160	572.23 ± 19.73 ^c^	1.27 ± 0.06 ^c^
2.5 mM PUT	0	602.83 ± 60.98 ^a^***	1.71 ± 0.08 ^a^***
	80	813.57 ± 7.81 ^b^**	2 ± 0.05 ^b^
	160	871.68 ± 21.45 ^b^***	2.02 ± 0.04 ^b^***
2.5 mM SPM	0	563.84 ± 75 ^a^***	1.31 ± 0.05 ^a^***
	80	867.32 ± 21.43 ^b^**	2.25 ± 0.04 ^b^***
	160	913.12 ± 32.42 ^b^***	2.29 ± 0.05 ^b^***
2.5 mM SPD	0	700.67 ± 56.78 ^a^	1.97 ± 0.01 ^a^*
	80	839.24 ± 19.65 ^b^**	1.94 ± 0.03 ^a^
	160	847 ± 24.13 ^b^***	2.02 ± 0.02 ^a^***

**Table 8 plants-12-01855-t008:** Calcium level in shoots and roots of tomato plants. PUT = putrescine; SPM = spermine; SPD = spermidine. Data are expressed as mean ± SE (*n* = 9). Mean values in the column marked by different letters are significantly different within the same group (*p* < 0.05; ANOVA and Tukey–Kramer test). Significant differences to CTRL are reported as *** *p* < 0.001.

Priming Solutions	NaCl (mM)	Shoots (μg Ca^2+^ ⋅ mg f.w.^−1^)	Roots (μg Ca^2+^ ⋅ mg f.w.^−1^)
CTRL	0	1.107 ± 0.034 ^a^	0.331 ± 0.053 ^a^
	80	0.899 ± 0.065 ^b^	0.427 ± 0.059 ^a^
	160	0.734 ± 0.113 ^c^	0.751 ± 0.050 ^b^
2.5 mM PUT	0	1.347 ± 0.126 ^a^	0.170 ± 0.007 ^a^ ***
	80	1.624 ± 0.024 ^b^ ***	0.217 ± 0.041 ^a^ ***
	160	1.804 ± 0.064 ^c^ ***	0.208 ± 0.023 ^a^ ***
2.5 mM SPM	0	1.392 ± 0.136 ^a^	0.142 ± 0.002 ^a^ ***
	80	1.371 ± 0.146 ^a^ ***	0.175 ± 0.031 ^a^ ***
	160	1.390 ± 0.151 ^a^ ***	0.121 ± 0.002 ^a^ ***
2.5 mM SPD	0	1.201 ± 0.056 ^a^	0.126 ± 0.001 ^a^ ***
	80	1.691 ± 0.066 ^b^ ***	0.155 ± 0.020 ^a^ ***
	160	1.647 ± 0.061 ^b^ ***	0.178 ± 0.015 ^a^ ***

## Data Availability

Not applicable.

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
