# Peer review of "Role of Polyamines in the Response to Salt Stress of Tomato"

_plants, 2023, doi:10.3390/plants12091855_

Round 1

Reviewer 1 Report

Dear Editor

Thank you very much for your invitation to review this manuscript) plants-2340788(:

Role of Polyamines in the Response to Salt Stress of Tomato

The work introduces very important information about the effect of putrescine, spermine and spermidine on tomato plants under salt stress conditions. The manuscript is good written, the introduction and the results are adequate, the discussion is very interesting and sufficient.

·         There are some comments highlighted in the pdf version, please see the pdf version

Best regards

Author Response

We thank the referee for the comments and suggestions. We made the changes and corrections as suggested. All the changes have been highlighted

Reviewer 2 Report

General comment:

Authors present a comprehensive analysis of the effect of seed priming with polyamines in combination with acclimation on irrigation with saline water in a cherry tomato variety, sensitive to salinity and based on morphological, physiological and biochemical parameters. The results are sound and conclusive. The introduction focuses on presenting the general problem of drought, salinity, and saline irrigation. In my opinion, in the introduction, authors should include  more information on previous knowledge on seed priming, acclimation to salinity and polyamine effects in plant performance.

Detailed comments:

Abstract: L21 – I propose: the acclimation consisted in sawing and growing seedlings for two weeks on non-saline soil with non-saline irrigation.

Introduction:

The introduction should stress more on acclimation techniques, especially for salt tolerance, techniques of seed priming and polyamines and stress reaction.

Materials and Methods:

L550-552: preliminary experiments for the identification of priming solutions should be detailed.

L567-573: please mention the number of plants per treatment – was this performed on pots with 5 seedlings?

Results:

L157-159: Although these are supplementary data (Fig. S1), the method should be explained in MM or figure legend. This would be an experiment without acclimation.

L181: I propose to add: …exposed or not to salt after acclimation.

L215-218: I propose to add that no significant differences were found between the levels of salinity.

L324-327: This sentence not clear. What means “scheme 11 and 18%”?

Fig. 13: Can the significance levels be calculated?

L348: a short introduction to TGase function should be added.

Discussion:

L423: Please correct:…in contrast to another study. In the reference mentioned (33) -where similar concentrations of all polyamines used?

L433: I would suggest: While the priming treatment with PAs improved the tolerance of tomato plants to salt stress with respect to chlorophyll maintenance.

L436: Please correct:…the increase in the content of sugars

Author Response

We thank the referee for the comments and suggestions. The replies point by point are reported below

Detailed comments:

Abstract:

L21 – I propose: the acclimation consisted in sawing and growing seedlings for two weeks on non-saline soil with non-saline irrigation.

REPLY: the phrase was rewritten (lines 21-23).

Introduction:

The introduction should stress more on acclimation techniques, especially for salt tolerance, techniques of seed priming and polyamines and stress reaction.

REPLY: more information has been added (lines 117-129).

Materials and Methods:

L550-552: preliminary experiments for the identification of priming solutions should be detailed.

REPLY: the information has been added (lines 565-569).

L567-573: please mention the number of plants per treatment – was this performed on pots with 5 seedlings?

REPLY: the information has been added (line 581).  

Results:

L157-159: Although these are supplementary data (Fig. S1), the method should be explained in MM or figure legend. This would be an experiment without acclimation.

REPLY: the information has been added in the legend of the fig. S1.  

L181: I propose to add: …exposed or not to salt after acclimation.

REPLY: done (line 190).

L215-218: I propose to add that no significant differences were found between the levels of salinity.

REPLY: the suggestion has been considered (lines 227-228)

L324-327: This sentence not clear. What means “scheme 11 and 18%”?

REPLY: unfortunately the following sentence was cancelled during the uploading: “SOD activity showed a different trend respect to POD and PPO: the most important results were observed with PUT priming (lines 334-335)”

Fig. 13: Can the significance levels be calculated?

REPLY: since this is a gel of activity performed under native conditions, it is not possible to determine the significance.

L348: a short introduction to TGase function should be added.

REPLY: done (lines 359-360) 

Discussion:

L423: Please correct:…in contrast to another study. In the reference mentioned (33) -where similar concentrations of all polyamines used?

REPLY: we reported a different citation where the authors used 1 mM PUT (line 437)

L433: I would suggest: While the priming treatment with PAs improved the tolerance of tomato plants to salt stress with respect to chlorophyll maintenance.

REPLY: done (lines 445-447)

L436: Please correct:…the increase in the content of sugars

REPLY: done (line 450)

Reviewer 3 Report

Research questions are well defined and within the aims and the scope of the journal. The introduction is adequate and includes in suitable way the relevant earlier publications. Materials are almost properly described. Methods are also almost properly described and used in a way that is possible to replicate. The investigation is performed to good technical standards. It is no ethical problem involved. A nicely conducted research (although a bit complicated) with conclusions well supported by the results. However, the level of English is inadequate. In some parts the text is not easy to follow. Moreover, the selection of the plant species test needs to justified. Also please use uniform letter fonts.

Author Response

We thank the referee for the suggestions that have been taken into consideration